# DCA-YOLOv8: A Novel Framework Combined with AICI Loss Function for Coronary Artery Stenosis Detection

**DOI:** 10.3390/s24248134

**Published:** 2024-12-20

**Authors:** Hualin Duan, Sanli Yi, Yanyou Ren

**Affiliations:** 1School of Information Engineering and Automation, Kunming University of Science and Technology, Kunming 650500, China; 20222104102@stu.kust.edu.cn (H.D.); 20232204107@stu.kust.edu.cn (Y.R.); 2Key Laboratory of Computer Technology Application of Yunnan Province, Kunming 650500, China

**Keywords:** stenosis detection, coronary artery, framework, preprocessing, attention

## Abstract

Coronary artery stenosis detection remains a challenging task due to the complex vascular structure, poor quality of imaging pictures, poor vessel contouring caused by breathing artifacts and stenotic lesions that often appear in a small region of the image. In order to improve the accuracy and efficiency of detection, a new deep-learning technique based on a coronary artery stenosis detection framework (DCA-YOLOv8) is proposed in this paper. The framework consists of a histogram equalization and canny edge detection preprocessing (HEC) enhancement module, a double coordinate attention (DCA) feature extraction module and an output module that combines a newly designed loss function, named adaptive inner-CIoU (AICI). This new framework is called DCA-YOLOv8. The experimental results show that the DCA-YOLOv8 framework performs better than existing object detection algorithms in coronary artery stenosis detection, with precision, recall, F1-score and mean average precision (mAP) at 96.62%, 95.06%, 95.83% and 97.6%, respectively. In addition, the framework performs better in the classification task, with accuracy at 93.2%, precision at 92.94%, recall at 93.5% and F1-score at 93.22%. Despite the limitations of data volume and labeled data, the proposed framework is valuable in applications for assisting the cardiac team in making decisions by using coronary angiography results.

## 1. Introduction

Cardiovascular disease is one of the leading causes of death worldwide, claiming approximately 17.9 million lives each year [1]. Coronary artery disease (CAD) is one of the most common and fatal cardiovascular diseases worldwide. The main cause of CAD is the accumulation of atherosclerotic plaques in the epicardial arteries [2], which leads to angina pectoris or heart attack. Accumulation of atherosclerotic plaque can lead to stenosis of the aortic lumen. Therefore, the detection of coronary artery stenosis is very important, and early detection of stenotic arteries allows early intervention and reduces mortality.

In recent years, deep-learning technology has made significant advancements in the medical field, especially in medical image analysis, which has pushed forward the application of computers in vascular stenosis detection. In coronary artery stenosis detection, physicians mainly rely on X-ray images to make their diagnosis. The deep-learning technology applied in this field initially segmented blood vessels from X-ray images, and physicians could then diagnose whether patients have stenosis based on the contour of blood vessels. For example, Yang et al. achieved a 91.7% F1-score for coronary vessel segmentation using a U-Net network [3]. Zhu et al. accomplished coronary vessel segmentation using a PSPNet network to segment the vessel contour to assist physician diagnosis [4]. Such algorithms are still semi-automated because they rely on physicians’ diagnoses. In fact, it makes sense for physicians to have a system that can automatically detect stenosis and support fully automated testing. One application of fully automated stenosis detection is the use of classification algorithms, where a computer model simply determines the presence of stenosis. For example, Jungiewicz et al. created a dataset using X-ray contrast images of 16 patients, including small fragment images of positive stenosis and small fragment images of non-stenosis negative for binary classification, with the innovative use of the vision-transformer classification network [5]. Ovalle-Magallanes et al. achieved an F1-score of 91.8% for classification accuracy on a smaller dataset by incorporating quantum computing into the neural network computation, using just 250 contrast images [6]. Another class of fully automated detection algorithms is object detection, which not only detects stenosis but also identifies the stenotic region. The latest research in object detection for coronary artery stenosis includes categorizing the diseased coronary artery into four types based on different patient perspectives, such as cranial (CRA) and left anterior oblique (LAO) views. These types are local stenosis (LS), diffuse stenosis (DS), bifurcation stenosis (BS) and chronic total occlusion (CTO). And the YOLOv5 detection model was utilized for diagnosis [7]. Additionally, the latest YOLOv8 detection model based on CNN networks was employed, combined with transfer learning techniques, to accomplish binary classification detection of coronary artery stenosis—whether there is stenosis on the vessel or not [8]. Of course, some of the latest technologies have been added to the object detection models, such as diffusion models, which, as a type of deep generative model, are widely used in computer vision. For instance, Li et al. proposed a quantum diffusion model with spatiotemporal feature sharing for real-time detection of stenosis, achieving an F1-score of 92.39%, which demonstrates significant performance [9]. Several other studies have also proposed relevant models for the detection of stenosis [10,11,12,13,14]. For example, Danilov et al. collected data from 100 patients, with a total of 8835 contrast images containing a clear annotation of the stenosis location inside each frame, and achieved a 96% F1-score and 94% mAP with the RFCN ResNet-101 V2 detection network [10]. Freitas et al. collected 132 frame images with significantly visible arterial stenosis from 50 patients and used a DeepCADD detection model to achieve a detection accuracy of 83% precision and 89.13% recall [11].

More meaningful to the physicians’ diagnosis is the object detection technology, which can determine whether stenosis is present on coronary angiography images and accurately determine the lesion area. However, when the object detection networks are performed on X-ray coronary angiography images, two problems exist: on the one hand, X-ray images contain noise and artifacts, particularly breathing artifacts that can blur blood vessel boundaries and mask blood vessel details, thereby affecting diagnostic results; on the other hand, stenotic regions of blood vessels often appear as small lesions, making the detection of these small targets more challenging for deep-learning algorithms. Therefore, more effective stenosis detection requires the network to have the following characteristics. First, in the contrast image, the structure of blood vessels is spread throughout the image, and the small branches of blood vessels have blurred contours, resulting in the complexity of the vessel structure in the contrast image. Therefore, a network with a strong ability to extract features is required to extract rich vessel structures. Second, X-ray contrast images contain more noise, with breathing artifacts and interference from other organs and tissues; therefore, the network must have anti-interference ability. Third, stenosis is formed because of the accumulation of some atherosclerotic plaques, which often manifest in a small area, and our network also needs to excel at detecting small targets.

With the progress of deep-learning techniques, an increasing number of efficient object detection networks have been proposed. Early object detection algorithms were mainly two-stage detection algorithms [15,16,17], such as the spatial pyramid pooling network (SPP-Net) [9], the region-based convolutional neural network (R-CNN) [16] and the Faster R-CNN [17]. Two-stage algorithms are performed by two different neural network branches: one generates the candidate box region, and the other classifies and detects the candidate box, which makes the overall algorithm more complex. The network of the single-stage algorithm [18,19,20] provides the prediction results along with the candidate frames, which significantly improves the inference speed and detection efficiency of the network. For example, SDD [18], RetinaNet [19] and YOLO [20] provide a qualitative leap in the detection speed. The early single-stage algorithms did not achieve the desired results in terms of detection accuracy because of their simple model structure. With the development of deep learning, single-stage algorithms have been proposed to solve this problem. For example, YOLOv5 [21] used an anchor-based and C3 module as the main extraction framework of the network during feature extraction and achieved 64.1% mAP50 on the COCO dataset. YOLOX [22] used the SimOTA label assignment strategy and the decoupled head prediction head, which achieved higher accuracy in a shorter training time. In addition, because of the excellent performance of the transformer [23] model in the field of natural language processing, Carion [24] and others introduced the model into the field of computer vision and proposed the DETR detection model. However, it has drawbacks, such as a large number of model parameters, a large number of training samples and difficulty in the convergence of training. Lv et al. optimized the training strategy and designed the RT-DETRv2 detection model by setting different numbers of sampling points for features at different scales in the deformable attention mechanism, based on the DETR detection model [25]. Duan et al. designed a bottom-up object detection method and developed the CenterNet++ detection model, which detects each object as a triplet of keypoints [26].

The YOLO family of network models uses diverse convolutional structures to enhance the feature extraction. These designs enable the models to capture and learn complex features in images more efficiently. YOLOv8 [27] has a diverse convolutional structure containing gradient streams enriched by the C2f module to enrich the model by connecting more branches across the layers, and the spatial pyramid pooling fast (SPPF) module captures information from different feature layers. Even so, for our coronary angiography images, the YOLOv8 network is still deficient because of the complexity of the vessel structure, interference of noise and artifacts. Some preprocessing methods for medical images can effectively improve network accuracy. Zhu et al. used the 3D Canny edge detection algorithm in MRI images of brain tumors to enhance the edge information of the lesion tissue [28]. Saifullah et al. achieved better segmentation results using particle swarm optimization with histogram equalization preprocessing [29]. In medical images, owing to the presence of background noise interference and breathing artifacts, attention mechanisms are often added to the network to enhance its immunity to interference. Ovalle-Magallanes et al. demonstrated that the model focuses more on the lesion area after adding the CBAM attention mechanism using visualization techniques [30]. In stenosis detection, the stenotic lesion area often appears in a small segment of a vessel, as shown in Figure 1. Coronary artery stenosis is difficult to detect accurately owing to the detection of small targets. For the detection of small targets, Wang et al. added two additional layers of feature maps to the detection head of the model to enhance the accuracy of small target detection in road vehicle detection [31].

In the field of coronary artery detection, the key issues that need to be addressed for more effective detection include the following: first, the framework must be able to effectively extract features from complex coronary angiography images, requiring the framework to have a strong feature extraction capability, which is essential for accurate medical image analysis. Second, the framework must possess anti-interference capabilities to effectively mitigate the effects of noise and artifacts in coronary angiography images, thereby enhancing the reliability and accuracy of the diagnostic process. Finally, the framework must address the challenge of detecting small targets, enabling more precise localization of the lesion area and achieving superior diagnostic outcomes. The framework effectively extracts key blood vessel contour information from the image, combats noise interference and concentrates on features within the stenosis region. It also effectively addresses the challenge of detecting stenosis in small target regions, thereby significantly improving the detection accuracy. The main contributions of this study are as follows:(1)The DCA-YOLOv8 object detection framework was designed according to the characteristics of coronary angiography images. The framework can maximize the extraction and focus on the stenosis information for fast and accurate stenosis detection.(2)The HEC preprocessing module uses a histogram equalized image with contours extracted using the Canny edge detection algorithm, enhancing the vascular region’s features.(3)We employed a detection head incorporating the AICI loss function to detect stenosis and small targets, utilizing an auxiliary detection frame. This method accelerates the convergence of the framework, improves the accuracy and achieves optimal detection results.

The remainder of this paper is organized as follows. Section 2 explains the proposed method in detail. The experimental setup and findings are presented in Section 3. Experimental results and comparative evaluations are presented in Section 4. Finally, Section 5 discusses and Section 6 summarizes the tasks, respectively.

## 2. Methods

The proposed basic framework for coronary artery stenosis detection consists of three parts. The first part is fused into the network using the DCA attention mechanism to extract rich vessel features. The second part is a HEC preprocessing enhancement module that uses a combination of histogram equalization and Canny edge detection. Finally, we used the output module that combines the AICI loss function detection header to converge faster and more accurately to complete the final detection. The flowchart is shown in Figure 2.

### 2.1. DCA Feature Extraction Module

In our study, we use the YOLOv8 main frame network to feature extract the input image, as shown in Figure 3.

YOLOv8 has a Conv + Batch Normalization + SiLU (CBS) module, C2f module, SPPF feature extraction module and neck feature fusion module, which is capable of extracting rich features. However, when applying YOLOv8 to stenosis detection using coronary angiography images, its capabilities are still insufficient. Therefore, we added a newly designed DCA module to the YOLOv8 network. According to our experiments, placing the DCA module before the C2f module yields better results.

The addition of attention mechanisms to networks has proven to be effective, and common attention mechanisms [32,33,34,35] are used in the context of computer vision, such as squeeze and excitation (SE) [32], CBAM [33], efficient channel attention (ECA) [34] and coordinate attention (CA) [35]. In coronary angiography, the vessel contour is the most important information in the image, and information regarding the vessels must be extracted and redundant information filtered out. Traditional feature extraction attention mechanisms, such as CBAM and SE, utilize global average pooling, which overlooks the positional information of the stenosis in blood vessels in the image. Thus, we propose adding the DCA attention mechanism to the framework, which directs the framework to focus more on the positional information of the stenosis in blood vessels in the image. The DCA structure is shown in Figure 4. Our proposed attention mechanism does not change the input channel. The DCA module consists of two serial sub-attention modules and processes feature inputs 
$x_{c}$
, which are computed by the following formula:
(1)
$$z_{c}^{h}\left(h\right)=\frac{1}{w}\sum_{0\leq i<w}x_{c}\left(h,i\right)$$


(2)
$$f^{h}=\delta \left(F_{1}\left(z_{c}^{h}\right)\right)$$


(3)
$$g^{h}=\sigma \left(F_{2}\left(f^{h}\right)\right)$$


(4)
$$f\left(x\right)=x\times \frac{ReLU6\left(x+3\right)}{6},\;ReLU6=\min(ReLU,\;6)$$


(5)
$$F’=x_{c}\times g^{h}$$


In Equation (1), 
$x_{c}\in R^{H\times W\times C}$
 are all pixel points of the input features, where 
h
 and 
w
 denote the height and width of the image, respectively. After Equation (1), the features are pooled in the wide direction dimension, preserving the position information in the h-direction: 
$z_{c}^{h}\in R^{H\times 1\times C}$
. After processing the important information in the h-direction using Equation (2), the vascular information in the h-direction is activated, which is represented as 
$f^{h}{\in R}^{H\times 1\times C/r}$
. In Equation (2), 
$F_{1}$
 represents the dimensionality reduction process using a 1 × 1 convolution kernel, with a scaling factor of r, and 
δ
 represents the activation function, which is expressed in Equation (4). It can exhibit a smoother gradient. The 
$g^{h}{\in R}^{H\times 1\times C}$
 attention vector in the h-direction is calculated by Equation (3), where 
$F_{2}\;$
 represents the upsampling process using a 1 × 1 convolution kernel to expand to the original channel dimension, and 
σ
 represents the sigmoid activation function. Finally, in Equation (5), the 
$g^{h}$
 attention vector is multiplied by the input to obtain the initial result. The second sub-module has the same process as the first one, which consists of pooling in the wide direction to maximize the retention of information in the w-direction position to obtain the final result output.

### 2.2. HEC Preprocessing Enhancement Module

In coronary angiography images, X-ray contrast imaging can reveal artifacts, blurred vessel contours and noise interference from other tissues, which makes it difficult to observe the vessels and results in poor contrast with the image background. Inspired by these two articles [28,29], we preprocess and enhance the image before inputting it into the framework to make blood vessel tissues more prominent, allowing stenotic regions to be more easily distinguished before being input into the DCA-YOLOv8 framework architecture.

The HEC module consists of adding pixel values from the histogram equalized image and the Canny edge extracted image. It can enhance the vascular feature area in the contrast image, thereby making it easier to distinguish stenosis in blood vessels. Histogram equalization is a method of transforming an original image to obtain a new image with a uniformly distributed grayscale levels. It widens the gray levels where there are many pixels and reduces them where there are few, balancing the original distribution of pixel values across a range of values. The image becomes brighter, and the contrast is enhanced. Histogram equalization is used to stretch the image non-linearly and redistribute the pixel values of the image to achieve a clear image. The per-pixel point mapping is as follows:
(6)
$$S_{K}=\left(\sum_{j=0}^{k}n_{j}\right)\times \frac{L}{N}$$


In Equation (6), L denotes the total gray level, which represents the converted gray level, and N denotes the number of pixels in the image. 
$n_{j}\;$
 represents the number of pixels included in the j-th gray level. 
$S_{K}$
 represents the converted pixel value. We used histogram equalization to enhance the vascular structure of our image, which made it easier to distinguish the vascular structure from the background. The transformed image is shown in Figure 5i.

The goal of the Canny edge detection operator is to find an optimal edge detection algorithm that applies Gaussian filtering to smooth the image with the aim of removing noise. It employs a dual-threshold approach to identify potential edges. The coronary angiography image is processed by the Canny operator to obtain a contour map of the vascular structure. Consequently, the stenotic region is included in the extracted contour, and this processing effectively enhances the features of the vessel contour and significantly boosts the features of the stenotic lesion. The transformed image is shown in Figure 5ii.

Finally, we combined the histogram equalized image with the Canny edge extraction results to obtain the final enhanced image, as shown in Figure 5iii. The HEC-processed image, which is more representative of our blood vessel features, was input into the framework for training.

### 2.3. Output

After extraction by the DCA-YOLOv8 framework backbone, three feature layers were obtained, which were 80 × 80, 40 × 40 and 20 × 20 in size. As depicted in Figure 2, the three output feature layers are concatenated together and simultaneously fed into the box subnet and the class subnet to perform box location and classifier. The final prediction result was obtained through the output module. In YOLOv8, the CIoU loss functions are employed for box location loss, and binary cross-entropy (BCE) is used for classification loss. In the framework we proposed, DCA-YOLOv8, we use AICI as the box location loss and BCE as the classification loss. The traditional loss function for detecting small targets suffers from a lack of accuracy in the fitted localization frame and slow convergence. We designed the AICI loss function according to the characteristics of coronary artery stenosis, which is more suitable for the detection and bounding box regression of small stenosis targets. The use of the AICI loss function can more accurately locate small areas of the stenosis.

In the YOLOv8 network, localization loss uses the CIoU loss function, which is composed of an IoU loss function and a penalty term. The calculation of the CIoU loss function is illustrated in Figure 6. The specific calculations were as follows:
(7)
$$L_{CIoU}=1-IoU+\frac{\rho^{2}(b,b^{gt})}{c^{2}}+\alpha \upsilon$$


(8)
$$\alpha =\frac{\upsilon }{(1-IoU)+\upsilon }$$


(9)
$$\upsilon =\frac{4}{\pi^{2}}{\left(arctan\frac{w^{gt}}{h^{gt}}-arctan\frac{w}{h}\right)}^{2}$$


In Equation (7), 
$\frac{\rho^{2}(b,b^{gt})}{c^{2}}$
 denotes the square of the distance between the target box and the center of the predicted box; 
$c^{2}$
 denotes the diagonal length of the smallest enclosing box that covers both boxes; and d represents the distance between the centers of two rectangular boxes. In addition, α is the weight function, which is computed based on 
υ
, and 
υ
 is used to measure the similarity of the aspect ratios. In the DCA-YOLOv8 framework, the CIoU loss function was improved by us.

For smaller target samples, the loss function can converge by using a larger auxiliary enclosing box. Setting a reasonable ratio value using the inner-IoU can accelerate convergence and improve the accuracy [36]. The inner-IoU is defined as follows: the ground truth (GT) box and anchor are denoted as 
$b^{gt}$
 and *b*, respectively, as shown in Figure 7. In Figure 7, the target box is the ground truth box, and the anchor box is the predicted box. The centers of the GT box and the inner-GT box are denoted by (
${x_{c}}^{gt},\;\;{y_{c}}^{gt}$
), while (
$x_{c}$
,
${\;y}_{c}$
) denotes the centers of the anchor and the inner anchor. The width and height of the GT box are denoted as 
$w^{gt}$
 and 
$h^{gt}$
, respectively, while the width and height of the anchor are represented by 
w
 and 
h
. The inner-IoU is calculated as follows:
(10)
$$b_{l}^{\mathrm{gt}}=x_{c}^{gt}-\frac{w^{gt}\times ratio}{2},\;b_{r}^{gt}=x_{c}^{gt}+\frac{w^{gt}\times ratio}{2}$$


(11)
$$b_{t}^{gt}=y_{c}^{gt}-\frac{h^{gt}\times ratio}{2},\;b_{b}^{gt}=y_{c}^{gt}+\frac{h^{gt}\times ratio}{2}$$


(12)
$$b_{l}=x_{c}-\frac{w\times ratio}{2},\;b_{r}=x_{c}+\frac{w\times ratio}{2}$$


(13)
$$b_{t}=y_{c}-\frac{h\times ratio}{2},\;b_{b}=y_{c}+\frac{h\times ratio}{2}$$


(14)
$$\mathrm{inter}=(\min(b_{r}^{gt},b_{r})-\max(b_{l}^{gt},b_{l}))\times (\min(b_{b}^{\mathrm{gt}},b_{b})-max(b_{t}^{\mathrm{gt}}{,\;b}_{t}))$$


(15)
$$\mathrm{union}=(w^{\mathrm{gt}}\times h^{\mathrm{gt}})\times ratio^{2}+(w\times h)\times ratio^{2}-\mathrm{inter}$$


(16)
$${IoU}^{inner}=\frac{inner}{union}$$


In inner-IoU, the corresponding scale factor ratio is set to control the scale size of the auxiliary bounding box. However, a change in the value of the ratio results in a change in the corresponding IoU value. The 
$\frac{\rho^{2}(b,b^{gt})}{c^{2}}$
 and 
αυ 
 additional term penalties in CIoU also change. Therefore, the penalty term factor in CIoU becomes inaccurate owing to the transformation of the ratio value. When inner-IoU is used, the area of the inner-GT anchor changes accordingly. Although these two loss terms were designed for the original IoU, now that we use the inner-IoU, the original loss terms have become inaccurate. Thus, we regulate the inaccuracy owing to the transformed IoU by adding two parameters 
ψ
 and 
τ
, which can be learned adaptively. Hence, our AICI loss function is defined as: 
(17)
$$L_{AICI}=1-IoU^{\mathrm{inner}}+\tau \frac{\rho^{2}\;(b,b^{gt})}{c^{2}}+\alpha \upsilon \psi$$


The AICI loss function is the box loss function used in the DCA-YOLOv8 framework proposed by us. In Equation (17), 
$\frac{\rho^{2}(b,b^{gt})}{c^{2}}$
 denotes the square of the distance between the target box and the center of the prediction box, and 
$c^{2}$
 denotes the diagonal length of the smallest enclosing box covering the two boxes. α is the weight parameter, and 
υ
 is used to measure the similarity of the aspect ratios. 
ψ
 and 
τ
 are the two parameters we added for adaptive learning. By setting the AICI loss function to a ratio value greater than one, the auxiliary frame is made larger, increasing the number of matches between the prediction frame and the gold-standard frame, thus reducing the problem of the vanishing gradient when the IoU is zero. This also increases the gradient value when the IoU is small, which makes the predicted box fit more closely to the target box. The AICI loss function enables the predicted frames to be more accurate, and the use of an auxiliary ratio value accelerates the convergence of the training, which leads to a higher level of accuracy.

## 3. Experimental Section

### 3.1. Datasets

Dataset I

This is a dataset for coronary artery stenosis classification. The coronary angiogram images are divided into many small patches, with those containing stenosis referred to as positive examples. The public dataset for stenosis classification introduced by Antczak and Łukasz [37] was used for evaluation. The original data of this dataset consist of 125 positive samples and 1394 negative samples, and because of the imbalance of positive and negative samples, the authors used a method to generate a dataset of 10k images, equally divided into stenosis and non-stenosis, with 5 k images of each. Deep-learning models require larger samples to train out more realistic results, and we used the generated dataset to train and verify the classification ability of our framework on real data.

Dataset II

This is a dataset for coronary artery stenosis object detection. One hundred patients from the Institute of Complex Problems of Cardiovascular Diseases [10] in Kemerovo, Russia, who were confirmed to have single-branch coronary artery disease, underwent coronary angiography using Siemens Coroscop and GEHealthcare’s Innova devices. The study was annotated by the local hospital, and it was annotated by three specialized physicians with more than 10 years of experience. The patients provided written informed consent, and coronary angiography images were retrospectively collected and processed. A total of 8325 grayscale images ranging in size from 512 × 512 to 1000 × 1000 pixels were selected, of which 80% were used for training, 10% for validation and 10% for testing. Each image had detailed standards for the stenotic regions. Non-random data segmentation ensures separate subsets for validation and testing, prevents bias in performance metrics and allows multiple experts to jointly perform data annotation.

### 3.2. Experimental Details

In the experiment, we set the input image size to 640 × 640. In the preprocessing stage, the Gaussian filter window size was set to 5 × 5, and the low and high thresholds were 10, 35, respectively. In the experiment, we used mosaic enhancement, which was stopped for the last 10 epochs. We used the Adam optimizer, with a learning rate of 0.002. The inner-CIoU auxiliary anchor ratio was set to 1.1. In the experiment, 
ψ
 and 
τ
 were initialized with the same values of 0.5. And the number of epochs was set to 100. The experimental platform framework was a Linux system, Pytoch = version 1.11.0, Python = 3.8, Cuda = 11.3, and the CPU was 16 vCPU Intel(R) Xeon(R) Platinum 8352V CPU @ 2.10 GHz. The model was run on a Ubuntu 20.04 operating system with NVIDIA GTX 4090.

### 3.3. Evaluation of Indicators

The evaluation refers to the precision (Prec), recall (Rec), F1-score (F1) and mean average precision (mAP), which can be calculated by the following formula:
(18)
$$Prec=\frac{TP}{TP+FP}$$


(19)
$$Rec=\frac{TP}{TP+FP}$$


(20)
$$F1=2\times \frac{Prec\times Rec}{Prec+Rec}$$


To clarify, true positives (TPs) refer to the number of cases where the intersection over union (IoU) between the detection results and the ground truth exceeds the threshold; false positives (FPs) refer to the number of cases where this IoU is below the threshold; and false negatives (FNs) refer to the number of stenoses that were not detected. This threshold of 0.5 is a common choice in stenosis detection, as referenced in [10,38]. Thus, in this experiment, the mean average precision (mAP) serves as the average accuracy metric for evaluating the model’s performance. The mAP is calculated by averaging precision across various recall rates, and a higher mAP value indicates superior detection performance by the model.

## 4. Results

In this section, there are four sets of experiments. The first set of experiments are a series of comparison experiments that demonstrate why we chose the HEC, DCA, and AICI modules. The second set of experiments, which are a series of ablation experiments, verify the roles of each module in the framework, including the HEC preprocessing module, DCA attention mechanism and AICI loss function. In the third set of experiments, DCA-YOLOv8 is compared with other object detection algorithms for coronary artery detection. The fourth set of experiments compare DCA-YOLOv8 with other authors’ frameworks for coronary artery classification and detection.

### 4.1. Comparative Experiment on the Function of HEC, DCA and AICI Modules

#### 4.1.1. Preprocessing Module Selection: Why Do We Choose the HEC Module?

In the preprocessing experiment, the main objective was to determine a more suitable preprocessing method for the enhancement of vessel region features. For this set of experiments, we used histogram equalization, Gamma transform, contrast-limited adaptive histogram equalization (CLAHE) and Canny edge detection to explore the most appropriate preprocessing method. Gamma transformation and CLAHE transformation images can be seen in Figure 8. The original image is first enhanced using histogram equalization, which enhances the visual effect and details of the image by redistributing the pixel intensity values of the image, such that the pixel values in the image are evenly distributed over the entire range of gray levels and then inputted into the YOLOv8 network. Subsequently, we augmented the image in the same way using Gamma transform, a gamma non-linear transform that allows the information of the blood vessel region to be enhanced. Then, we augmented the image in the same way using CLAHE, a contrast-limited adaptive histogram equalization method. Our aim is to find a more effective preprocessing method that can increase the edge information of blood vessels effectively. We use Canny edge detection to extract the contours of the blood vessels and then sum the pixels with the enhanced image, making the vessel contours more distinct in the original image and enhancing the blood vessel information. The experimental results are listed in Table 1.

In Table 1, we can see that using HE results in better performance than using CLAHE. The reasons are as follows: (1) In coronary angiography images, the contrast between the vessels and the background is not very large, and in such cases, traditional HE performs better. (2) The background in coronary angiography images is simple, with only two states: the vessels and the background. Using a global technique better distinguishes the difference between the two and also avoids the issue of using local techniques, where small regions with only background information may lead to wasted image data. And the Gamma transformation cannot effectively increase the contrast between blood vessels and the background. Canny edge extraction can effectively extract the vessel edge information and enhance the vessel details, which helps our framework better identify the stenotic lesion areas.

#### 4.1.2. Comparative Experiment of Adding Different Attention Mechanisms to the YOLOv8 Backbone

The purpose of this set of experiments is to demonstrate the effectiveness of our DCA module design combined with YOLOv8. We compared it with each of the four currently popular attention mechanisms added to the YOLOv8 backbone: CBAM, ECA, SE and CA [32,33,34,35]. Details of the data are presented in Table 2.

From the experimental results shown in Table 2, we propose that the DCA mechanism combined with YOLOv8 showed that the F1-scores were 1.26%, 0.22%, 1.4% and 0.7% higher than those of CBAM, ECA, SE and CA, respectively. The mAP values were 1.21%, 0.8%, 0.08% and 0.34% higher than those of CBAM, ECA, SE and CA, respectively.

#### 4.1.3. Experiments Comparing Different Loss Functions in DCA-YOLOv8

The purpose of this set of experiments is to verify the effectiveness of the proposed AICI loss function. We compare the improved AICI loss function with two sets of experiments, the CIoU loss function and the inner-CIoU loss function, to demonstrate that our proposed AICI has faster convergence and higher accuracy. We also compared the number of epochs required when the mAP accuracy reaches above 90%. The fewer the number of epochs, the faster the convergence of the loss function.

In the experimental results in Table 3, the framework incorporating the AICI loss function achieved the best experimental results, reaching 95.83% of the F1-score and 97.6% of the mAP. In Table 3, we can see that the AICI loss function achieves a mAP of over 90% with only 66 epochs, which also proves that the AICI loss function converges faster compared to other loss functions. As shown in Figure 9, with the AICI loss function, the convergence was faster, and a smaller box loss was reached for the same number of epochs.

### 4.2. Ablation Experiments Between HEC, DCA and AICI Modules

We demonstrate the impact of the HEC preprocessing module, the DCA feature extraction module and the output module combined with AICI on the performance of our framework through ablation experiments. Details of the data are presented in Table 4. The baseline is the YOLOv8 network, and the experimental results show that the HEC preprocessing module and the DCA feature extraction module also achieve satisfactory results when used together. The best results were achieved when using the DCA, HEC and output modules combined with the AICI loss function: 95.83% F1-score and 97.6% mAP. The detailed analysis is presented in the discussion of Section 5, and we present the visualization of the ablation experiments in Figure 10. The green box represents the final result of the framework’s detection of stenosis in Figure 10.

Both the DCA feature extraction module and the HEC preprocessing enhancement module are designed to enhance the framework’s focus on the feature information of stenotic vessels. We combined the DCA and HEC modules in the ablation experiments for comparison with the previous network. In the visualized results shown in Figure 10(a2,b2,a3,b3), it is evident that after incorporating the DCA and HEC preprocessing enhancement modules, the framework effectively suppresses the appearance of false positive results. It can be seen in (a1) and (b1) that with the addition of the DCA and HEC modules, the framework can focus more on the stenotic region and reduce the number of true positive misses.

Figure 11 shows the ablation experiments with and without the AICI loss function for the framework. The green box represents the final result of the framework’s detection of stenosis. Comparing the experimental results in (a1)–(a3) and (b1)–(b3), when the AICI loss function is used, there is no redundant information regarding the vessel in the stenosis detection box. The target location and size are more suitable for stenosis labeling, making it easier for physicians to observe.

### 4.3. Comparison with Other Representative Object Detection Models in Stenosis Detection (Dataset II)

We compare our method with the currently popular object detection algorithms for the stenosis detection dataset. We choose representative object detection networks, such as Yolov5 [21], Yolov7 [39], DETR [24], Faster-Rcnn [17] and RT-DETRv2 [25] for comparison. And the experimental results are shown in Table 5. At the same time, we also compare the size of each model parameter.

In this set of experiments, to ensure a fair comparison, we set the epoch number for all models to 100 and used default settings for all other parameters. From the experimental results, it is evident that our DCA-YOLOv8 framework outperforms the others, achieving an F1-score of 95.83% and a mAP value of 97.6%, both of which exceed those of the other four model types. At the same time, our model has the smallest number of parameters, with a size of 3.24 M, which is significantly smaller than that of other models. Faster-Rcnn, YOLOv7 and RT-DETRv2 are commonly used basic models, and their performance in medical image recognition is poor. This is because they are designed for natural images and do not make improvements tailored to the characteristics of coronary angiography images. This is the reason why our framework can achieve the best results.

### 4.4. Comparison with Other Classification and Object Detection Models

#### 4.4.1. Comparison with Other Authors’ Classification Models

Our proposed DCA-YOLOv8 framework suppresses the detection output head, which is essentially a classification model, as shown in Figure 2, suppressing the box subnet. The quality of the classification results can reflect the strength of our framework’s ability to extract features from coronary angiography images. This set of experiments demonstrates that our framework possesses better feature extraction ability for blood vessels. We used publicly available categorized datasets (Dataset I). We proved that the framework is multitasking by suppressing the detection branch of the proposed network and allowing our framework to be experimented on Dataset I using only the classifiers. Several authors [6,37,40] used Dataset I for the classification. Antczak et al. trained on an artificial dataset and then fine-tuned it on a real dataset, achieving 90% accuracy [37]. Ovalle-Magallanes et al. added quantum computation to the output layer of the network and achieved 94% accuracy on Dataset I [6]. Gil-Rios et al. used a feature selection module in a network to achieve 87% accuracy on Dataset I [40].

As can be seen in Table 6, DCA-YOLOv8 also shows an excellent level of stenosis categorization, with an accuracy of 93.2%, which is 3.2% higher than that of Karol Antczak and 6.2% higher than that of Miguel-Angel Gil-Rios. In these models, our framework achieved the best recall rate.

#### 4.4.2. Comparison with Other Authors’ Detection Models in Coronary-Artery-Related Research

We found relevant information from other authors on Dataset II for coronary artery stenosis detection. Danilov et al. achieved good results using the RFCN ResNet-101 V2 detection network on Dataset II [10]. To ensure fairness in the comparison, we reproduced Danilov‘s method in our experimental setup, and the results are shown in Table 7. We reproduced Han’s ResNet50 network as a baseline, combined with proposal-shifted spatiotemporal tokenization (PSSTT) and transformer-based feature aggregation (TFA) models, and we conducted experiments on the stenosis detection dataset (Dataset II) [14]. Cosmo et al. achieved an F1-score of 87.92% in Dataset II. His method involved using two datasets to perform federated learning and subsequent validation on Dataset II. Because the authors used another undisclosed dataset, we cannot reproduce the results. The comparison results are as follows.

Table 7 shows the comparison with other methods. As can be seen in Table 7, our results outperform the current best model (RFCN ResNet-101 V2) by 0.99% in the F1-score and 2.92% in mAP. This is because the backbone of RFCN ResNet-101 V2 does not have as strong feature extraction capabilities as our framework, and our use of DCA attention mechanism can resist interference from other noise. Of course, we also employed the HEC preprocessing enhancement module, which makes it easier for our framework to identify the diseased and stenotic areas. Our framework achieved the highest mAP, outperforming the best model by 2.92%.

## 5. Discussion

### 5.1. How to Select the Best-Performing Enhancement Module

In the first set of experiments, we experimentally verified the effectiveness of the HEC module by selecting different preprocessing methods: Gamma transform, histogram equalization, CLAHE and Canny edge extraction. In Table 1, we can see that using HE results in better performance than using CLAHE. The reasons are as follows. (1) In coronary angiography images, the contrast between the vessels and the background is not very large, and in such cases, traditional HE performs better. (2) The background in coronary angiography images is simple, with only two states: the vessels and the background. Using a global technique better distinguishes the difference between the two and also avoids the issue of using local techniques, where small regions with only background information may lead to wasted image data. And the Gamma transformation cannot effectively increase the contrast between blood vessels and the background. From the experimental results in Table 1, it can be seen that using histogram equalization and Canny edge extraction as preprocessing methods for the framework is effective. This combination is the most effective preprocessing method for enhancing coronary angiography images. Compared with the original image, the F1-score improved by 0.91%, and the mAP improved by 0.68%. From the experimental results in Table 2, it can be seen that our designed DCA feature extraction module outperformed commonly used attention mechanism modules in detecting coronary artery stenosis. The DCA attention mechanism focuses more on the positional information of vessels in the image, including that of stenotic lesions, making it more sensitive to vessel stenoses. The experimental results in Table 3 show the effect of different loss functions on the model. In Figure 9, the box loss plot clearly shows the difference between the three loss functions: the Inner-CIoU loss function converges faster than CIoU. However, the final loss values achieved by Inner-CIoU and CIoU are not significantly different. When using the AICI loss function, the loss value decreases more rapidly, indicating a faster convergence speed and the achievement of a lower loss value.

### 5.2. Ablation Experiments for Each Module

In the second group of experiments, the functional role of each module was verified through ablation experiments. The HEC preprocessing enhancement module’s role is to enhance the vascular feature information of the coronary arteries in the images, making it easier to distinguish vascular information from background features and simultaneously enhancing the feature information of stenotic lesions within the vessels. As shown in Table 4, the recall rate significantly improved after the addition of the HEC preprocessing enhancement module, indicating that coronary stenosis was more easily detected, and the false negative rate was reduced. Figure 10 shows that when the HEC+DCA module was used, the framework more easily detected stenotic lesions and suppressed false positive judgments. Figure 11 shows that using the detection head with the AICI loss function enables the framework’s localization frame to more accurately pinpoint stenotic lesions, with less redundant information in the localization frame, thereby increasing the accuracy of detection results. Furthermore, the experimental results in Table 4 demonstrate that the F1-score and mAP of the framework improved after employing the AICI loss function, thereby validating its effectiveness.

### 5.3. Analyzing the Reasons Why the Model Achieved Better Results

In the third set of experiments, a comparison with other object detection methods showed that DCA-YOLOv8 achieved the best results. The main reasons for this are as follows:(1) The YOLOv8 network was used as the basis of our framework and was capable of extracting rich features. (2) We used the HEC preprocessing module, which combines histogram equalization and Canny edge detection to effectively enhance the contours and details of the blood vessels, making it easier for the framework to detect stenosis in the blood vessels. (3) We integrated the DCA attention mechanism into the framework, enhancing its focus on information of the stenosis areas. (4) We used a detection head incorporating the AICI loss function along with a larger auxiliary detection frame during the training process, which accelerated framework convergence and improved the detection of small targets with stenosis blood vessels.

### 5.4. Comparison with Current Methods

In the fourth set of experiments, we suppressed the detection branch of the output coordinate frame, making the framework categorical. The testing was performed on a publicly available stenosis dataset, and the results are shown in Table 5. In contrast to several other studies [17,21,24,25,39], we achieved the best recall, with 93.5% of the experimental results. This set of experiments demonstrates the competitiveness of our framework for the classification of coronary stenosis. We also compared our method with other authors’ stenosis detection methods, with the results shown in Table 7. Cosmo et al. [41] used federated contrast learning, achieving an F1-score of 87.92% and proving the effectiveness of federated learning, but they did not achieve satisfactory results in terms of detection accuracy. Danilov et al. [7] used the deeper ResNet101 [42] as the network extraction backbone and achieved a good F1-score, but other specific parameters were not given. Our proposed DCA-YOLOv8 framework shows potential for practical use, achieving the highest mAP value of 97.6% for stenosis detection and a high F1-score of 95.83%. Our results outperform the current best model (RFCN ResNet-101 V2) by 0.99% in the F1-score and 2.92% in mAP. This is because we designed a HEC preprocessing enhancement module, a DCA attention mechanism and a new loss function (AICI) suitable for small targets based on the characteristics of CAD images, which enables our DCA-YOLOv8 framework to achieve excellent inspection results.

## 6. Conclusions

Here, we proposed a new framework for the detection of coronary stenosis. This includes preprocessing, a feature extraction network and a detection head. First, in the preprocessing stage, we designed a HEC enhancement module tailored to the characteristics of coronary angiography images, which increases the contrast between vessels and the background, enabling the framework to more accurately identify stenosis lesions. Second, in the feature extraction module, we incorporated the DCA attention mechanism, which directs the framework to focus more on the vessel region information and the characteristics of stenosis lesions, thereby improving the framework’s accuracy. Finally, for the detection of small targets in stenosis, we designed the AICI loss function, which accelerates the convergence during framework training and enhances the framework’s accuracy. The experimental results show that our proposed framework achieved precision, recall, F1-score and mAP values of 96.62%, 95.06%, 95.83% and 97.6%, respectively. The average precision score reached a peak of 97.6%, and all other metrics were also highly competitive.

The advantages of this study are as follows. (1) The framework proposed in this paper achieves more accurate detection of coronary artery stenosis, thereby effectively assisting physicians in diagnosing the condition. (2) The novel DCA-YOLOv8 framework incorporates innovative improvements for coronary artery stenosis detection, demonstrating higher accuracy than other networks. The proposed stenosis detection framework could better assist physicians in rapidly and accurately identifying the narrowed coronary artery stenosis areas.

Our method has certain limitations. Due to the dataset, the framework we propose can only detect the presence of stenosis on coronary angiography images and cannot further classify the types of stenosis or identify arteries with severe blockage. The detection results from our framework can only assist physicians in determining whether stenosis is present. Further treatment decisions, such as stent placement and other interventions, must be made by physicians based on the degree of stenosis and the patient’s health condition. Future work will focus on utilizing more annotated coronary angiography images to enable the model to assess the type of stenotic lesions and determine the degree of stenosis, providing physicians with more valuable supplementary reference. This will further enhance the performance of the framework and broaden its applicability.

## Figures and Tables

**Figure 1 sensors-24-08134-f001:**
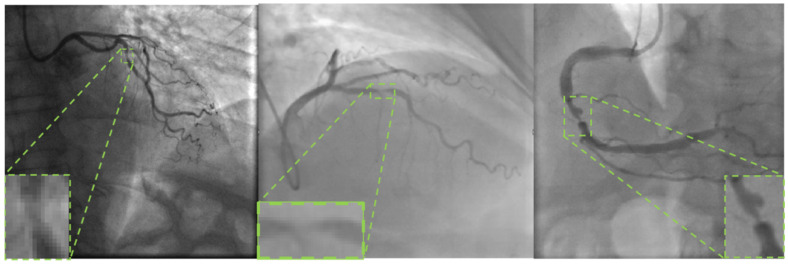
Example of coronary artery stenosis detection.

**Figure 2 sensors-24-08134-f002:**
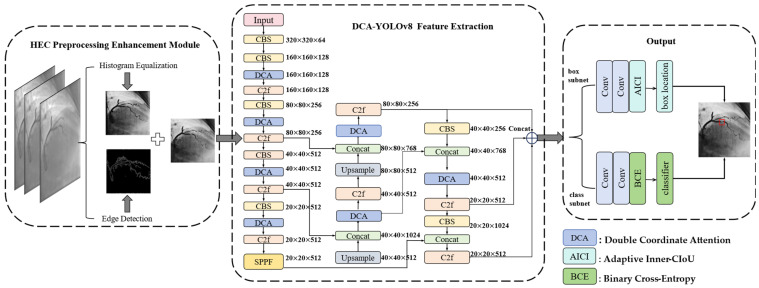
DCA-YOLOv8 overall structural frame diagram.

**Figure 3 sensors-24-08134-f003:**
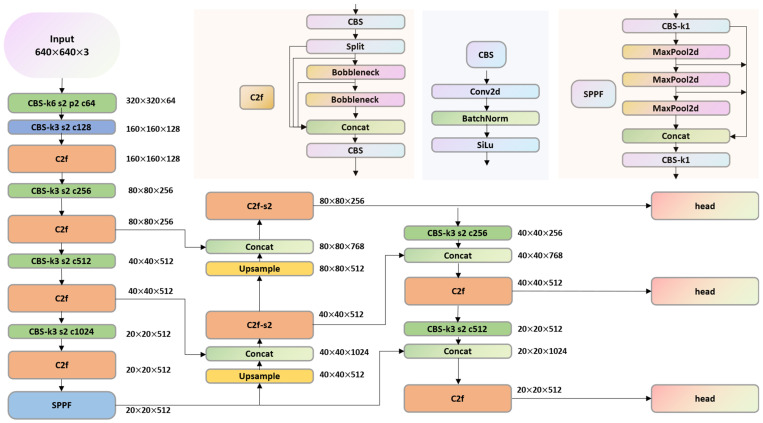
Schematic diagram of YOLOv8 structure.

**Figure 4 sensors-24-08134-f004:**
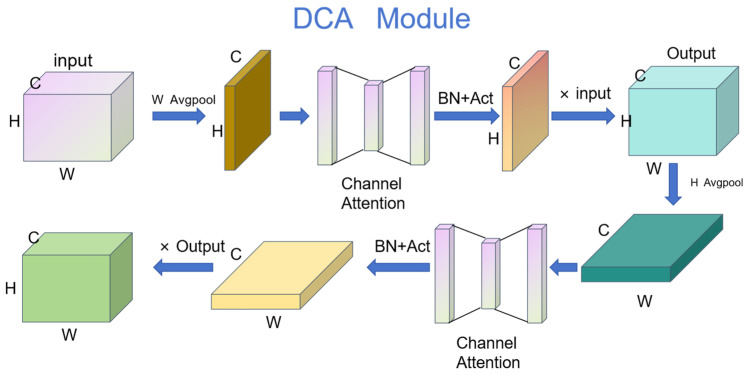
Schematic diagram of DCA module structure.

**Figure 5 sensors-24-08134-f005:**
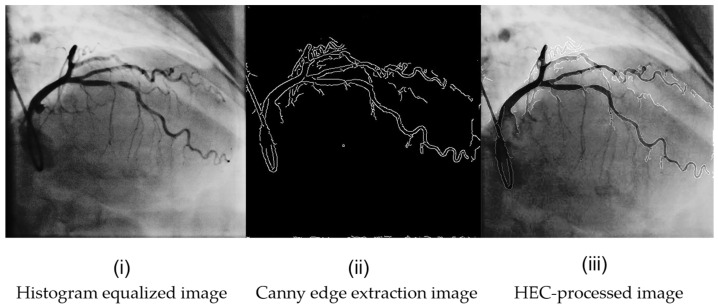
Preprocessing enhancement effect diagram: (**i**) Histogram equalized image, (**ii**) Canny edge extraction image, (**iii**) HEC-processed image.

**Figure 6 sensors-24-08134-f006:**
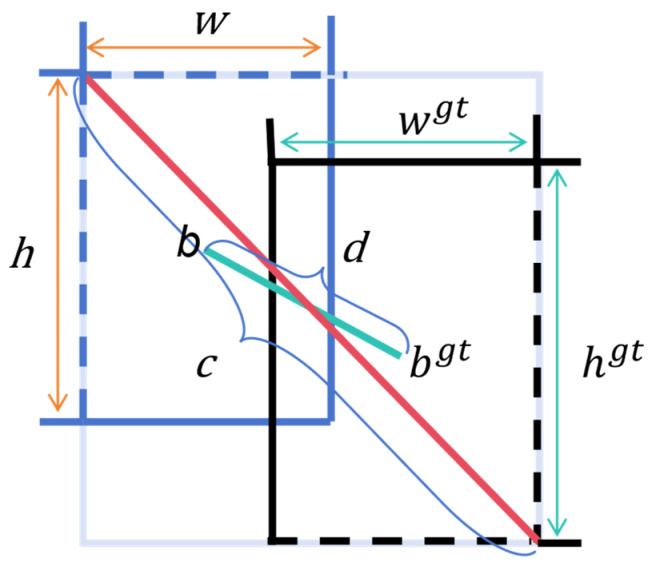
CIoU loss function calculation.

**Figure 7 sensors-24-08134-f007:**
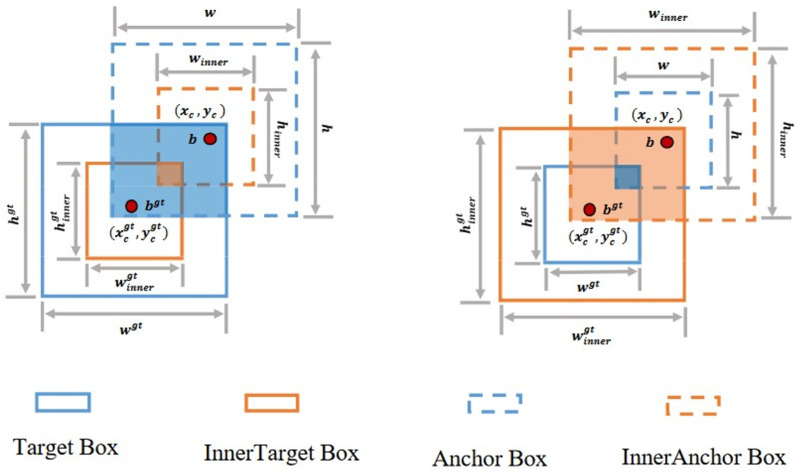
Schematic diagram of inner-IoU structure.

**Figure 8 sensors-24-08134-f008:**
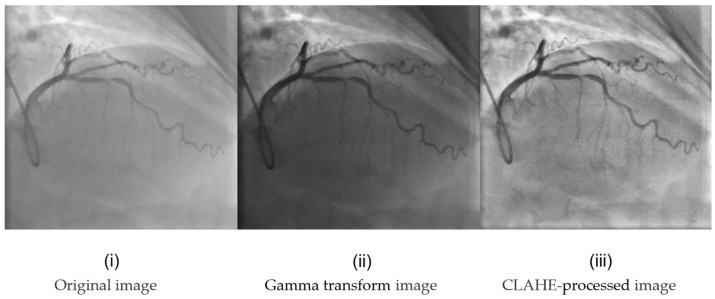
Gamma transformation and CLAHE-processed visualization images. (**i**) Original image, (**ii**) Gamma transform image, (**iii**) CLAHE-processed image.

**Figure 9 sensors-24-08134-f009:**
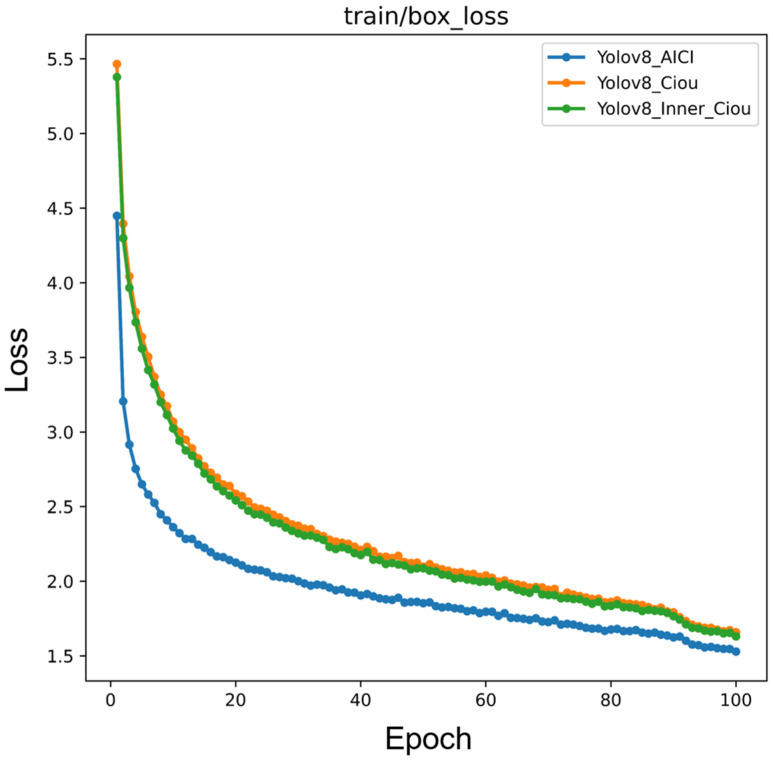
Box loss regression with three different loss functions.

**Figure 10 sensors-24-08134-f010:**
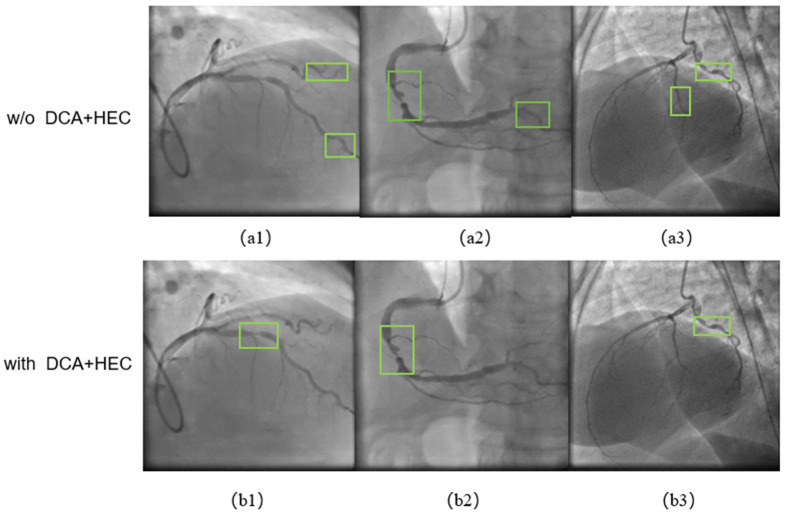
Coronary angiography image detection results without DCA and HEC modules in (**a1**–**a3**), with DCA and HEC modules in (**b1**–**b3**). The green box represents the final result of the framework’s detection of stenosis.

**Figure 11 sensors-24-08134-f011:**
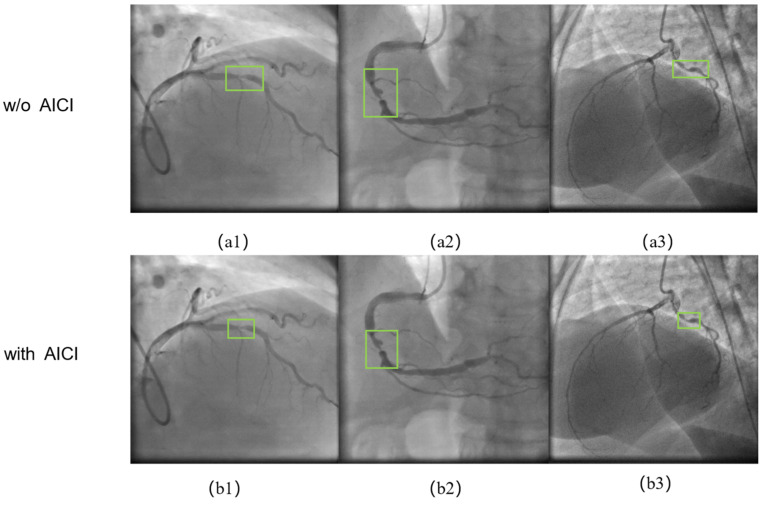
Ablation experiments with or without the use of the AICI loss function in coronary stenosis detection. Results of assays without the AICI loss function in (**a1**–**a3**) and results of assays with the AICI loss function in (**b1**–**b3**). The green box represents the final result of the framework’s detection of stenosis.

**Table 1 sensors-24-08134-t001:** Preprocessing selection experiment.

Processing Method	Prec (%)	Rec (%)	F1 (%)	mAP (%)
OI ^1^	95.13	92.72	93.91	95.35
HE ^2^	95.25	93.83	94.53	95.67
CLAHE ^3^	95.36	92.96	94.34	95.08
GT ^4^	93.83	91.42	92.61	93.86
CEOI ^5^	95.56	93.46	94.5	95.72
HEC ^6^	95.67	93.98	94.82	96.03

^1^ Original image. ^2^ Histogram equalization. ^3^ Contrast-limited adaptive histogram equalization. ^4^ Gamma transform. ^5^ Canny edge extraction and original image. ^6^ Canny edge extraction and histogram equalization.

**Table 2 sensors-24-08134-t002:** Comparative experiments with different attention mechanisms in DCA-YOLOv8.

Method	Prec (%)	Rec (%)	F1 (%)	mAP (%)
CBAM	95.06	93.82	94.44	95.21
ECA	96.10	94.00	95.04	95.62
SE	94.92	93.08	93.99	96.34
CA	95.62	93.56	94.58	96.08
Ours (DCA)	96.32	94.86	95.58	96.42

**Table 3 sensors-24-08134-t003:** Comparative experiments with different loss functions.

Loss Function	Prec (%)	Rec (%)	F1 (%)	mAP (%)	Epoch
CIoU	96.32	94.86	95.58	96.42	86
Inner-CIoU	96.46	94.67	95.56	96.48	78
Ours (AICI)	96.61	95.06	95.83	97.60	66

**Table 4 sensors-24-08134-t004:** Results of ablation experiments for each module.

Method	Prec (%)	Rec (%)	F1 (%)	mAP (%)
w/o DCA w/o HEC w/o AICI *	95.13	92.72	93.91	95.35
w/o HEC w/o AICI	95.73	93.64	94.67	95.86
w/o DCA w/o AICI	95.67	93.98	94.82	96.03
w/o AICI	96.32	94.86	95.58	96.42
Ours	96.61	95.06	95.83	97.60

* w/o means “without”.

**Table 5 sensors-24-08134-t005:** Comparison with other object detection networks.

Method	Prec (%)	Rec (%)	F1 (%)	mAP (%)	Parameter
Faster-Rcnn	93.53	92.57	92.97	94.86	23.51 M
YOLOv5	95.42	92.12	93.74	95.82	7.32 M
YOLOv7	95.13	92.72	93.91	95.35	5.92 M
DETR	76.56	74.26	78.39	-	41.38 M
RT-DETRv2	96.71	94.56	95.61	96.53	23.51 M
Ours	96.62	95.06	95.83	97.6	3.24 M

**Table 6 sensors-24-08134-t006:** Comparison with other authors’ classification models.

Authors	Acc (%)	Pre (%)	Rec (%)	F1-Score (%)
Antczak et al. [37]	90	-	-	-
Ovalle et al. [6]	94	95	92	94
Gil-Rios et al. [40]	87	-	-	-
Ours	93.2	92.94	93.5	93.22

**Table 7 sensors-24-08134-t007:** Comparison with other authors’ detection models in the detection of coronary artery stenosis.

Method	Prec (%)	Rec (%)	F1 (%)	mAP (%)
Danilov et al. [10]	95.23	94.38	94.84	94.68
Cosmo et al. [41]	92.91	83.43	87.92	-
Tan et al. [14]	87.56	84.83	86.18	85.32
Ours	96.62	95.06	95.83	97.6

## Data Availability

The data underlying the results presented in this study are available in Dataset I [37] (https://github.com/KarolAntczak/DeepStenosisDetection/tree/master/datasets). Accessed on 1 November 2023. The data underlying the results presented in this study are available in Dataset II [10] (https://data.mendeley.com/datasets/ydrm75xywg/1). Accessed on 12 November 2023. The code is available at https://github.com/duanhualin/DCA-YOLOv8.git (accessed on 1 November 2023).

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
