# Peer review of "DCA-YOLOv8: A Novel Framework Combined with AICI Loss Function for Coronary Artery Stenosis Detection"

_sensors, 2024, doi:10.3390/s24248134_

Round 1
Reviewer 1 Report
Comments and Suggestions for Authors
The manuscript presents a promising framework for coronary stenosis detection, introducing innovative modules such as the HEC preprocessing and the DCA attention mechanism. While the reported results are compelling, there are several areas where additional detail and contextualization could further enhance the manuscript’s impact and clarity.
One critical area for improvement is the discussion of the state-of-the-art in coronary angiography detection. While the manuscript mentions related work, it lacks a detailed and systematic review of recent advancements, particularly in object detection methods and their application to coronary stenosis detection. Including a discussion of benchmarks such as the ARCADE challenge and other relevant public datasets would provide a stronger foundation for understanding the novelty and contributions of the proposed approach. Additionally, it would be useful to discuss how the proposed method compares with the state-of-the-art in terms of methodology, dataset characteristics, and performance metrics.
In terms of methodology, the HEC preprocessing module effectively combines histogram equalization and Canny edge detection to enhance vascular features. While this approach is innovative, it remains unclear how specific it is to coronary angiography versus other medical imaging tasks. The authors could strengthen this section by comparing the results of the HEC module with alternative preprocessing techniques, such as CLAHE (Contrast Limited Adaptive Histogram Equalization) or adaptive filtering methods.
The clinical implications of the results also require further elaboration. While the manuscript reports high precision and recall, these metrics alone do not provide a complete picture of the framework’s clinical utility. For example, how do the model’s predictions align with the diagnostic processes of cardiologists? Are false negatives, which might miss critical stenosis, or false positives, which could lead to unnecessary interventions, more concerning in this context? Including case studies or visual examples that illustrate the model’s performance in clinically significant cases, such as borderline stenosis or heavily occluded arteries, would add practical relevance to the findings.
Regarding experimental comparisons, the fairness of the evaluation is a notable concern. The models being compared were tested on different datasets, making direct comparisons challenging. If it is not feasible to evaluate all models on the same dataset, the authors should provide a detailed discussion of how the datasets differ in size, quality, and clinical diversity. This would help readers better understand the limitations of such comparisons. Benchmarking the framework against publicly available datasets would also enhance the transparency and reproducibility of the findings.
The focus on detecting small stenotic regions is a strength of the proposed framework, particularly as this is an area where traditional methods often struggle. The AICI loss function and the DCA module are highlighted as key components in achieving this performance. However, the manuscript could provide more concrete evidence to support this claim, such as visual examples or detailed quantitative metrics illustrating the improvement in detecting small lesions. It would also be valuable to examine how the framework’s performance varies across different lesion sizes and whether this variation could impact its clinical utility.
Author Response
Point 1: One critical area for improvement is the discussion of the state-of-the-art in coronary angiography detection. While the manuscript mentions related work, it lacks a detailed and systematic review of recent advancements, particularly in object detection methods and their application to coronary stenosis detection. Including a discussion of benchmarks such as the ARCADE challenge and other relevant public datasets would provide a stronger foundation for understanding the novelty and contributions of the proposed approach. Additionally, it would be useful to discuss how the proposed method compares with the state-of-the-art in terms of methodology, dataset characteristics, and performance metrics.
Response 1: Thank you very much. According to your professional commends: (1) we have added a detailed and systematic review on recent developments, and added References [7-9],[12-14], [25-26] in the second and fourth paragraphs of the introduction. Furthermore, we have added a set of experiments comparing the latest object detection technique, such as [25] in subsection 4.5 of the revised version. (2) for the dataset of ARCADE, as far as we know, there only 3 research papers using it in object detection task, and as they only using parts of data of this dataset, their results can not be used for the comparison in this paper. (3) we have added a detailed comparison in terms of methodology, dataset characteristics and performance metrics in section 3 and section 5 (discussion part).
Point 2: In terms of methodology, the HEC preprocessing module effectively combines histogram equalization and Canny edge detection to enhance vascular features. While this approach is innovative, it remains unclear how specific it is to coronary angiography versus other medical imaging tasks. The authors could strengthen this section by comparing the results of the HEC module with alternative preprocessing techniques, such as CLAHE (Contrast Limited Adaptive Histogram Equalization) or adaptive filtering methods.
Response 2: Thank you very much for your professional work. We have added a set of experiments comparing the CLAHE transformation with our HEC module. The experimental results are added in Table 1 of section 4. And the detailed analysis of the better performance of HEC module is added in 2nd paragraph of section 4.1 and section 5.1.
Point 3: (1) The clinical implications of the results also require further elaboration. (2)While the manuscript reports high precision and recall, these metrics alone do not provide a complete picture of the framework’s clinical utility. For example, how do the model’s predictions align with the diagnostic processes of cardiologists? Are false negatives, which might miss critical stenosis, or false positives, which could lead to unnecessary interventions, more concerning in this context? (3) Including case studies or visual examples that illustrate the model’s performance in clinically significant cases, such as borderline stenosis or heavily occluded arteries, would add practical relevance to the findings.
Response 3: Thank you very much, according to your commends: (1) we have added the further elaboration of clinical implications of our work. (2) thank you for this meaningful and professional question. Firstly, as far as we know existing literature is based on the validation of model performance using existing datasets, which are reliable in their selection of data, and our work follows the same approach. Secondly, in case of using unreliable data which exist false negatives as you mentioned above, such problem can be attributed to the study field of the data’s reliability which is the next study point of our research. (3) the visual examples that illustrate the model’s performance are presented in fig 10. and fig 11.
Point 4: (1) Regarding experimental comparisons, the fairness of the evaluation is a notable concern. The models being compared were tested on different datasets, making direct comparisons challenging. If it is not feasible to evaluate all models on the same dataset, the authors should provide a detailed discussion of how the datasets differ in size, quality, and clinical diversity. This would help readers better understand the limitations of such comparisons. (2) Benchmarking the framework against publicly available datasets would also enhance the transparency and reproducibility of the findings.
Response 4: Thank you for your professional question. (1) Regarding the fairness of the comparison, our revision is as follow: Firstly, we have conduct a new experiment in which the model of [14] are performed on the same dataset as our model; Secondly, regarding [11], since there is no public procedure, it is impossible to verify its model, therefore we have excluded it from the comparison with this paper. Thirdly, The revised results are added in table 7 of the revised draft. (2) In order to enhance the transparency and reproducibility of our framework, We have added the link to the framework based on this dataset at the end of the article.
Point 5: (1) The focus on detecting small stenotic regions is a strength of the proposed framework, particularly as this is an area where traditional methods often struggle. The AICI loss function and the DCA module are highlighted as key components in achieving this performance. However, the manuscript could provide more concrete evidence to support this claim, such as visual examples or detailed quantitative metrics illustrating the improvement in detecting small lesions. (2) It would also be valuable to examine how the framework’s performance varies across different lesion sizes and whether this variation could impact its clinical utility.
Response 5: Thank you for your professional work. (1) We have provided more concrete evidence to support the claim you mentioned above, such as, the visual examples are added in fig10 and fig 11, more detailed quantitative metrics are added in table 3 of the revised draft, which can better illustrating the improvement in detecting small lesions. (2) Thank you for such meaningful advice. Since the current dataset does not support this type of research, we are unable to conduct such experiments for the time being. However, this is a research direction that interests us and is on our agenda for the future.

Reviewer 2 Report
Comments and Suggestions for Authors
The manuscript proposed a YOLOv8-based model for coronary artery stenosis detection. It is well-written and technically sound. However, some observations need to be addressed:
1. The quality of images, formulas, and equations is low. It needs to be improved.
2. The input and output dimensions of the DCA module in equations 1 to 5 are not clear. Please include each variable's dimension when writing the module's explanation.
3. It is not clear where the DCA module was set up. Please describe how many parameters this module adds to the model and how it differs from CBAM.
4. In equation 6, there is no explanation for $n_j$
5. Also, why use this global equalization technique instead of a local equalization (CLAHE)?
6. In Figure 5, please add more details in the caption.
7. The Gamma Transform was introduced in Section 4. Include the explanation and a visual example of this transformation in Section 2.
8. Include the number of parameters of the models in the comparison tables.
9. I found some research that can enhance the literature review and be helpful to verify and compare their performance: https://doi.org/10.1016/j.eswa.2023.120234, https://doi.org/10.1016/j.compbiomed.2023.106546, https://doi.org/10.3390/electronics11213570, https://doi.org/10.3389/fcvm.2023.944135, https://doi.org/10.1109/ICICAT57735.2023.10263760
Comments on the Quality of English LanguageThe quality of English is sufficient
Author Response
Point 1: The quality of images, formulas, and equations is low. It needs to be improved.
Response 1: Thank you for your professional work. We have checked the manuscript carefully and corrected.
Point 2: The input and output dimensions of the DCA module in equations 1 to 5 are not clear. Please include each variable's dimension when writing the module's explanation.
Response 2: Thank you for your careful work. We have added the dimension information for the input to output processes and provided a clearer explanation of Equations 1 to 5 in the manuscript.
Point 3: It is not clear where the DCA module was set up. Please describe how many parameters this module adds to the model and how it differs from CBAM.
Response 3: Thank you very much. According your professional commands, we have marked the position of the DCA module more clearly in Figure 2 of the revised version. Because the addition of the DCA attention mechanism resulted in a very small change in model parameters, approximately 0.02M, we did not delve into a detailed discussion of it in the paper. The difference between DCA and CBAM is: the pooling process of CBAM involves performing global max pooling and global average pooling along the channel dimension, followed by activation using shared parameters. In contrast, the pooling process of DCA first performs pooling and activation along the width dimension, and then pooling and activation along the height dimension. These are two distinct processes, and we also use different activation functions.
Point 4: In equation 6, there is no explanation for $n_j$.
Response 4: Thank you for your careful work. We have added a detailed explanation of the parameters in Equation 6.
Point 5: Also, why use this global equalization technique instead of a local equalization (CLAHE)?
Response 5: Thank you very much for your professional work, according to your question, we have added experimental evidence in the revised draft showing that using HE results in better performance than using CLAHE. And the reason is:(1) in coronary angiography images, the contrast between the vessels and the background is not very large, and in such cases, traditional HE performs better. (2) the background in coronary angiography images is simple, with only two states: vessels and background. Using a global technique better distinguishes the difference between the two and also avoids the issue of using local techniques, where small regions with only background information may lead to wasted image data.
Point 6: In Figure 5, please add more details in the caption.
Response 6: Thank you for your careful work. We have added detailed information in Figure 5.
Point 7: The Gamma Transform was introduced in Section 4. Include the explanation and a visual example of this transformation in Section 2.
Response 7: Thank you for your suggestion. We have added more explanation about Gamma Transform along with a visual demonstration in the subsection 4.1 of the revised manuscript.
Point 8: Include the number of parameters of the models in the comparison tables.
Response 8: Thank you for your professional suggestion. We have added the number of parameters of the models in table 5.
Point 9: I found some research that can enhance the literature review and be helpful to verify and compare theirperformance:https://doi.org/10.1016/j.eswa.2023.120234, https://doi.org/10.1016/j.compbiomed.2023.106546,https://doi.org/10.3390/electronics11213570, https://doi.org/10.3389/fcvm.2023.944135, https://doi.org/10.1109/ICICAT57735.2023.10263760
Response 9: Thank you very much. We have cited the following references in the [5], [8], [12-14].

Reviewer 3 Report
Comments and Suggestions for Authors
Comment to authors.
1) I strongly suggest to add a dictionary of acronyms, there are many in the text,
many are defined the first time they are used, however, some others appear in the text with no definition causing a lot of confusion.
2) Figure 6. The ground truth box is the target box ? The blue color box is the predicted box ? Where is "b" in Figure 6 (Line 246) ? What is "d" in Fig 6?
3) Figure 7. I consider the AICI loss function a contribution of your paper. However, Figure 7 is hard to visualize in detail. At a zoom value of 200% something can be read, and apparently the intuition says the word says "inner" but at 300% it says "tnnar", I am not sure.
4) Is the following extract correct ? The centers of the GT box and the Inner-GT box are denoted by (𝑥𝑐 𝑔𝑡,𝑦𝑐 𝑔𝑡), while (𝑥𝑐,𝑦𝑐) 254 denotes the centers of the anchor and the inner anchor, while the widths and heights of 255 the anchor are denoted by 𝑤𝑔𝑡and ℎ𝑔𝑡, respectively
5) Briefly Clarify Target box, ground truth box, anchor box, etc.
6) Blue box says "Targer" -> target
7) Line 268. How are the parameters learned by your algorithm ? I did not see any reference to these parameters again.
8) Line 123 - 141, What is "the framework" ? Your proposal ?
9) Line 136. I can hardly consider YOLOv8 as your contribution, then again "the framework" appears in the description. Please revise the english writing.
10) 2. Methods.
Figure 2, right hand side, the 2 vertical sequential boxes in the output subgraph, what is that?
11) Figure 3, I hope the editorial knows what to do but I had to switch to 300% zoom as to be able to read the block diagram. (it looks great, though.)
12) Subsection 2.2 HEC
Nice work, nice figures.
13) Subsection 2.3 Output. Please review all of it, see my comment (above)
14) Line 308 -> Adam optimizer
15) Subsection 3.3
Clearly there is an error, and perhaps an equation or two equations are missing.
16) 4. Rssults. OK, good experimental design.
17) 4.1.1 Preprocessing...
OK. Table 1 ahows the results, (Line 350), the conclusions or at least some discussion is necessary (to support the case of canny edge detection and historgram..,)
18) 4.1.2 Attention...
Chenge to present tense, why the narrative was change to past tense ?
From the experimental results in Table 2, we proposed that the DCA mechanism 360 combined with YOLOv8 showed that the F1-score were 1.26%, 0.22%, 1.4%, and 0.7% 361 Sensors 2024, 24, x FOR PEER REVIEW 11 of 19
higher than CBAM, ECA, SE, and CA, respectively. The mAP values were 1.21%, 0.8%, 362 0.08%, and 0.34% higher than those of CBAM, ECA, SE, and CA, respectively.
19) Line 406. Consider "Physicians" instead of "doctors"
20) Subsection 4.3 NO FURTHER COMMENTS,
5. DISCUSSION , GREAT SECTION
6. CONCLUSION, GOOD WORK.
Author Response
Point 1: I strongly suggest to add a dictionary of acronyms, there are many in the text,
many are defined the first time they are used, however, some others appear in the text with no definition causing a lot of confusion.
Response 1: Thank you for your professional suggestion. We have added "Abbreviations" at the end of the revised version, which lists the correspondence between abbreviations and their full names.
Point 2: Figure 6. The ground truth box is the target box? The blue color box is the predicted box? Where is "b" in Figure 6 (Line 246)? What is "d" in Fig 6?
Response 2: Thank you for your professional work. We have checked the manuscript carefully and corrected. The meaning of "b" is indicated in Figure 6, and the meaning of "d" is explained in detail in the manuscript.
Point 3: Figure 7. I consider the AICI loss function a contribution of your paper. However, Figure 7 is hard to visualize in detail. At a zoom value of 200% something can be read, and apparently the intuition says the word says "inner" but at 300% it says "tnnar", I am not sure.
Response 3: Thank you for your careful work. I apologize for the mistake; the correct term should be "inner." We have already corrected it in Figure 7 in the revised draft in Figure 7.
Point 4: Is the following extract correct? The centers of the GT box and the Inner-GT box are denoted by (?? ??, ?? ??), while (??, ??) denotes the centers of the anchor and the inner anchor, while the widths and heights of the anchor are denoted by ??? and ℎ??, respectively.
Response 4: Thank you for your professional work. That (?? ??, ?? ??) and (??, ??) denote the centers of the anchor and the inner anchor is correct. Since the transformation of the inner anchor involves scaling the rectangular anchor proportionally, the center points of the anchor and the inner anchor will not be transformed. the widths and heights of the anchor are denoted by ??? and ℎ??, respectively. We have already corrected the sentence 'While widths and heights of the anchor are denoted by ??? and ℎ??, respectively in line 290 of the manuscript.
Point 5: Briefly Clarify Target box, ground truth box, anchor box, etc.
Response 5: Thank you very much for your suggestion. We have provided a detailed and clear explanation regarding the Target box, ground truth box, anchor box in the fourth paragraph of section2.3.
Point 6: Blue box says "Targer" -> target.
Response 6: Thank you very much. We have already corrected it.
Point 7: Line 268. How are the parameters learned by your algorithm? I did not see any reference to these parameters again.
Response 7: Thank you for your careful work. The two parameters have been added to the DCA-YOLOv8 structure, and they can be learned automatically in the backpropagation during the learning process of our framework. In line 343 of the manuscript, we have provided the initial values for the two parameters."
Point 8: Line 123 - 141, What is "the framework"? Your proposal?
Response 8: Thank you for your careful questions. Yes,DCA-YOLOv8 is the framework that we proposed.
Point 9: Line 136. I can hardly consider YOLOv8 as your contribution, then again "the framework" appears in the description. Please revise the English writing.
Response 9: Thank you very much for your suggestion. We introduced the contributions of our proposed framework in subsection 2.3 and distinguished it from YOLOv8. And we have corrected the writing in English.
Point 10: Figure 2, right hand side, the 2 vertical sequential boxes in the output subgraph, what is that?
Response 10: I apologize for any confusion caused by our poor expression regarding our framework. To provide a clearer understanding, we have redrawn Figure 2 and added detailed explanations in section 2.3 of the revised draft.
Point 11: Figure 3, I hope the editorial knows what to do but I had to switch to 300% zoom as to be able to read the block diagram. (it looks great, though.)
Response 11: Thank you very much. As for Figure 3, we have enlarged it for better clarity. (We have revised it for better clarity.)
Point 12: Subsection 2.2 HEC
Nice work, nice figures.
Response 12: Thank you very much.
Point 13: Subsection 2.3 Output. Please review all of it, see my comment (above)
Response 13: Thank you for your professional work. Based on the comments mentioned above, we have reviewed Subsection 2.3 carefully and revised it.
Point 14: Line 308 -> Adam optimizer
Response 14: Thank you very much. We have corrected it in the revised draft.
Point 15: Subsection 3.3
Clearly there is an error, and perhaps an equation or two equations are missing.
Response 15: Thank you very much for your professional work. Due to our careless, there exists an error of the equations in subsection 3.3, and we have correct the formula which is marked as red.
Point 16: 4. Results. OK, good experimental design.
Response 16: Thank you very much for your encouragement.
Point 17: 4.1.1 Preprocessing...
- Table 1 allows the results, (Line 350), the conclusions or at least some discussion is necessary (to support the case of canny edge detection and historgram..,)
Response 17: Thank you very much for your suggestion. We have added a detailed discussion to support the case of Canny edge detection and histogram equalization in subsection 4.1.1 of the manuscript.
Point 18: 4.1.2 Attention...
Change to present tense, why the narrative was change to past tense ?
Response 18: Thank you very much for your careful work. We have already corrected tense in the manuscript.
Point 19: Line 406. Consider "Physicians" instead of "doctors"
Response 19: Thank you very much for suggestion. We have corrected the word to "Physicians" in Line 460.
Point 20: Subsection 4.3 NO FURTHER COMMENTS,
Response 20: Thank you for your professional work. We have added some comments in Section 4.3 regarding the comparative experiments, with further comments in Section 5.3.
Point 21: DISCUSSION, GREAT SECTION, CONCLUSION, GOOD WORK.
Response 21: Thank you very much for your recognition of these two sections.

Round 2
Reviewer 1 Report
Comments and Suggestions for Authors
The authors have addressed the comments and made efforts to improve the manuscript; however, significant editing is still required to enhance clarity and readability. In its current form, certain sections remain difficult to understand, which could hinder comprehension for readers.
For example, the authors use phrases like "which we acquired at the 520 Ospedali Riuniti of Ancona (Italy)" when referring to data or work that does not belong to them.
Overall, careful attention to language and structure is needed to ensure the manuscript communicates its findings better.
Extensive editing of the manuscript is required.
Author Response
Point 1: The authors have addressed the comments and made efforts to improve the manuscript; however, significant editing is still required to enhance clarity and readability. In its current form, certain sections remain difficult to understand, which could hinder comprehension for readers. For example, the authors use phrases like "which we acquired at the 520 Ospedali Riuniti of Ancona (Italy)" when referring to data or work that does not belong to them. Overall, careful attention to language and structure is needed to ensure the manuscript communicates its findings better.
Response 1: Thank you very much. We have conducted a detailed review and revision of the language and structure throughout the manuscript, with particular attention to the experimental section. For the datasets used in the experiments, we have added more detailed descriptions. For example, Dataset I is used for stenosis classification, while Dataset II is used for stenosis detection, which helps readers better understand them. In the experimental section, we have provided more detailed descriptions for the titles of each experiment. Additionally, we have included the corresponding dataset for each experiment, along with more specific details. Our summary of the experiments is presented at the beginning of Section 4 (Results), and carefully reviewing this part will greatly aid in understanding the content of the experimental section.

Reviewer 2 Report
Comments and Suggestions for Authors
Answering the comments improved the manuscript according to the previous review. Please verify the link to the proposed model's source code; it appears to be unavailable. It is desirable to add a GitHub repository.
Author Response
Point 1: Answering the comments improved the manuscript according to the previous review. Please verify the link to the proposed model's source code; it appears to be unavailable. It is desirable to add a GitHub repository.
Response 1: Thank you very much. We have put the new GitHub link in the abstract of the manuscript.
